# New-Generation Materials for Hydrogen Storage in Medium-Entropy Alloys

**DOI:** 10.3390/ma17122897

**Published:** 2024-06-13

**Authors:** Dagmara Varcholová, Katarína Kušnírová, Lenka Oroszová, Jens Möllmer, Marcus Lange, Katarína Gáborová, Branislav Buľko, Peter Demeter, Karel Saksl

**Affiliations:** 1Faculty of Materials, Metallurgy and Recycling, Technical University of Kosice, Letna 9, 042 00 Kosice, Slovakia; katarina.gaborova@tuke.sk (K.G.); branislav.bulko@tuke.sk (B.B.); peter.demeter@tuke.sk (P.D.); karel.saksl@tuke.sk (K.S.); 2Institute of Materials Research, Slovak Academy of Sciences, Watsonova 47, 040 01 Kosice, Slovakia; kkusnirova@saske.sk (K.K.); loroszova@saske.sk (L.O.); 3Institut für Nichtklassische Chemie, Permoserstraße 15, 04318 Leipzig, Germany; moellmer@inc.uni-leipzig.de (J.M.); lange@inc.uni-leipzig.de (M.L.)

**Keywords:** hydrogen, hydrogen storage, absorption, medium-entropy alloys, AlTiNbX

## Abstract

This study presents the design, preparation, and characterization of thirty new medium-entropy alloys (MEAs) in three systems: Al-Ti-Nb-Zr, Al-Ti-Nb-V, and Al-Ti-Nb-Hf. The hardness of the alloys ranged from 320 to 800 HV_0.3_. Among the alloys studied, Al_15_Ti_40_Nb_30_Zr_15_ exhibited the highest-reversible hydrogen storage capacity (1.03 wt.%), with an H/M value of 0.68, comparable to LaNi_5_, but with a reduced density (5.11 g·cm^−3^) and without rare earth elements. This study further reveals a strong correlation between hardness and hydrogen absorption/desorption; higher hardness is responsible for reduced hydrogen uptake. This finding highlights the interplay between a material’s properties and hydrogen storage behavior in MEAs, and has implications for the development of efficient hydrogen storage materials.

## 1. Introduction

Hydrogen is widely used as a process gas in industrial applications, such as reduction processes, ammonia production, oil refining, and chemical production, but it is also used, for example, in electrical engineering in the production of semiconductors, contact cleaning, etc. However, hydrogen is also an energy carrier, which means that it can be used to store, transfer, and release energy. Hydrogen plays an important role in the European Union’s (EU) energy and climate strategy [1]. The EU has set ambitious targets to achieve a sustainable, low-carbon, and competitive energy future, in which hydrogen is seen as a key part of this strategy. A significant feature of hydrogen is its environmental friendliness during its combustion with oxygen, enabling hydrogen energy to be harnessed without releasing CO_2_, a major contributor to air pollution. Thus, it offers a solution for the decarbonization of industrial processes and economic sectors in which a reduction in carbon emissions is urgent but still difficult to achieve. All of this makes hydrogen essential to meet the EU’s commitment to carbon neutrality by 2050. Currently, hydrogen, as an energy carrier, represents only a small fraction of the EU’s global energy mix, which the European community intends to significantly change in its strategy in the coming years. Its production, which today is almost exclusively from fossil fuels, especially from natural gas or coal, and which leads to the annual release of 70 to 100 million tons of CO_2_ in the EU, will undergo a significant change. In order for hydrogen to contribute to carbon neutrality, it is therefore necessary to significantly change the way it is initially obtained and, at the same time, to expand its use in industry and transport in order to decarbonize various sectors. As part of its strategy, the EU has set an ambitious target of producing at least 6 gigawatts (GW) of electrolyzers to produce hydrogen from renewable energy sources by 2024, and 40 GW of electrolyzers by 2030 [2]. These targets are intended to support the rapid development of the hydrogen economy and create the market conditions for hydrogen to be used a clean fuel and energy carrier. By achieving this EU goal, it is assumed that the share of hydrogen in Europe’s energy mix will increase from the current amount of less than 2% to 13–14% by 2050. In terms of use, the EU’s ambition is to replace fossil fuels with hydrogen in carbon-intensive industrial processes, such as the steel and chemical industries, reducing greenhouse gas emissions and further strengthening the global competitiveness of these industries [3,4].

The primary techniques for hydrogen storage consist of both physical and chemical methods. The physical approaches involve compressing hydrogen, turning it into a liquid, employing cryo-compression, or adsorbing it physically. Conversely, the chemical methods rely on metal hydrides, complex hydrides, liquid organic hydrides, and caster hydrides [5].

Metal hydrides originate from the reversible interaction between a metal or alloy capable of hydride formation, or an intermetallic (IMC) compound, and hydrogen gas (H_2_).

IMCs can encompass AB_5_ types (e.g., LaNi_5_, CaNi_5_) [6], AB2 varieties (e.g., ZrMn_2_, ZrV_2_, ZrCr_2_), AB types (e.g., TiFe [6]), and A_2_B compounds (e.g., Ti_2_Ni, Zr_2_Fe). In a binary IMC, A typically represents a hydride-forming element, while B denotes a transition or non-transition metal/element that does not typically form a stable hydride under standard conditions.

Most IMC hydrides exhibit reversible hydrogen storage capacities that are practically achievable and typically do not surpass 2 wt.% H. For instance, AB_5_ hydrides generally achieve around 1.5 wt.% H, AB_2_ hydrides about 1.8 wt.% H, and BCC alloys approximately 2.0 wt.% H. Consequently, extensive investigations have been conducted on light metal hydrides, like MgH_2_ [7] and AlH_3_, for hydrogen storage purposes. Despite their notable gravimetric capacities, reaching 7.6 wt.% H for MgH_2_ and 10.1 wt.% H for AlH_3_, these materials are beset by other limitations. For instance, MgH_2_ requires elevated working temperatures [8], while AlH_3_ synthesis directly necessitates kbar H_2_ pressures and high temperatures [9].

The identification of metal hydrides suitable for efficient hydrogen storage was proclaimed in the late 1960s, coinciding with the emergence of intermetallic hydrides of the AB_5_ configuration, where A represents a rare earth metal and B stands for Ni or Co. Notably, compounds like LaNi_5_ have demonstrated remarkable efficacy as chemically reversible reservoirs for hydrogen gas.

Solid-state hydrogen storage stands out as a highly promising method. Particularly noteworthy within this domain are alloys, which have emerged as leading materials for hydrogen storage due to their advantageous cost, safety features, and operational parameters, notably their high energy density per unit volume. For instance, a widely employed commercial hydrogen storage alloy found in nickel–metal hydride batteries is the AB_5_ alloy, characterized by a CaCu_5_ crystal structure. Nonetheless, conventional alloys encounter numerous challenges in hydrogen storage. Each alloy exhibits a unique set of strengths and weaknesses, and their collective performance still falls short of the targets set by the EU. However, a promising avenue lies in a novel class of alloy materials known as medium-entropy alloys (MEAs) [10].

Regarding complex hydrides, over the last twenty years, research endeavors have progressively directed attention towards solid-state materials incorporating complexes of light elements, such as boron, aluminum, and nitrogen. These complexes, including (BH_4_)^−^ [9], [AlH_4_]^−^ [8], (AlH_6_)^3−^, (NH_2_)^−^, and (NH)^2−^, are coordinated with one or more metals [11].

Among the various categories of multiprincipal element alloys (MPEAs/HEAs), there has been notable attention given to refractory alloys featuring a bcc lattice, along with substitutions involving lightweight elements.

Within the entire chain of hydrogen production, from renewable sources, to storage, to use in industry or in the conversion of fuel cells into electricity, this work focuses on the possible storage of hydrogen in metal alloys. In particular, the development of new alloys (medium-entropy alloys), which ensures high compression and volume density by binding hydrogen in the metal lattice, is the focus of this research activity. Alloys with moderate entropy values, ranging from 1.0 R to 1.5 R, have gained increasing interest among researchers [12,13]. This growing interest stems from their remarkable characteristics, including their impressive mechanical properties [13,14], strong corrosion resistance [15,16], and notable structural stability [17]. The mixing entropy effect in these alloys typically leads to the formation of single-phase structures, such as Face-Centered Cubic (FCC), Body-Centered Cubic (BCC), and Hexagonal Closed Packed (HCP) [18]. An “excellent” step forward in this direction is represented by the publication of M. Sahlberg et al.’s paper, entitled “Superior hydrogen storage in high entropy alloys” [18]. In this study, the authors investigated the hydrogenation of a high-entropy alloy in a TiVZrNbHf solid solution with a BCC structure, and found that it was possible to absorb an extremely large amount of hydrogen (2.7 wt.% hydrogen). The amount of hydrogen corresponded to an H/M ratio of 2.5, and set a world record for volumetric energy density of 219 kg H/m^3^ [16,19].

Certain systems, like TiVNbTa, TiVZrNb, and TiVZrNbHf, have exhibited phase separation phenomena during the cycling of hydrogen absorption and desorption [20,21], particularly under the elevated temperatures typically required for hydrogen desorption from hydrides [22].

In recent examinations, the absorption characteristics of hydrogen in the Zr-deficient TiVZrNb alloy, featuring a non-equimolar composition, have been scrutinized to enhance procedural synthesis and cycling performance. This alloy crystallizes into a solitary-phase bcc structure, manifesting hydrogen assimilation within a single progression, mirroring the behavior of the TiVZrNbHf alloy. The utmost hydrogen uptake that has been attained is roughly 1.75 H/M (2.5 wt.% H). The most efficient hydrogen cycling attributes are apparent in the bct hydride variation, demonstrating an analogous capacity (1.8 H/M) achieved via reactive milling. A consistent reversible capacity of approximately 2.0 wt.% H has been noted, and attributed to the lack of disproportionate or irreversible segregation throughout the hydrogenation process [23].

The most auspicious candidates for hydrogen storage consist of bcc MPEAs/HEAs derived from refractory metals, possessing a VEC ≥ 5.0, coupled with substantial lattice distortion and undergoing single-phase transitions throughout the hydrogenation process.

For instance, leveraging these findings, TiVCrNbH_8_ (VEC = 5) emerges as a viable candidate for solid-state hydrogen storage. This HEA-based hydride exhibits a reversible hydrogen storage capacity of 1.96 wt.% H at room temperature (RT), and moderate H_2_ pressures. Notably, it does not necessitate any intricate activation procedures for hydrogen absorption [24].

This research focuses on developing new alloys that offer both low density and high hydrogen storage capacity, without the utilization of rare earth elements.

## 2. Methodology and Experiments

Alloys Al_x_Ti_x_Nb_x_Zr_x_, Al_x_Ti_x_Nb_x_V_x_, and Al_x_Ti_x_Nb_x_Hf_x_ were produced from pure elements (Al-99.95%, ChemPur (Karlshure, Germany); Ti-99.95%, Alfa Aesar (Haverhill, MA, USA); Nb-99.8%, Alfa Aesar (Haverhill, MA, USA); Zr-99.8%, ChemPur (Karlshure, Germany); V-99.9% ChemPur (Karlshure, Germany); Hf-99.9%, ChemPur (Karlshure, Germany)) by arc melting in a Mini Arc Melting System MAM-1 (Edmund-Bűhler, Bodelshausen, Germany) furnace in an atmosphere of protective gas (Ar with purity 99.999%) [25]. To ensure a clean atmosphere for melting the alloys, a small quantity of titanium was melted in the furnace chamber using the guttering method. This process effectively removed impurities and gas contaminants from the furnace environment. The arc melting of the charge was performed 5 times, with the sample being rotated after each melting. This ensured the homogeneity of the elements in the volume of the alloy.

In the first step, the density of the alloys was determined by the Archimedes method, using precise laboratory scales Kern ABT 120-4M (Merck Life Science spol. s r.o, Bratislava, Slovakia) with a special adapter ABT-A01 (Merck Life Science spol. s r.o, Bratislava, Slovakia) for density measurement.

From the bulk (button-shape) alloys, metallographic cuts were prepared using the standard method of casting in resin, followed by grinding and polishing.

The chemical composition of the prepared alloys was subsequently determined using energy-dispersive X-ray spectroscopy (EDX) with a Tescan Vega 3 LM scanning electron microscope (Tescan, Brno, The Czech Republic) [26].

Microhardness tests HV_0.3_ were performed on the polished surface using a Wilson-Wolper Tukon 1102 hardness tester (Berg Engineering & Sales Company, Inc., Rolling Meadows, IL, USA), equipped with the Vicker’s type of microindenter. Ten indentations were made during the microhardness tests, and the mean value and standard deviation were calculated from the measurements.

The determination of nanoindentation hardness and elastic modulus were performed by measuring the nanoindentation using a Nano Indenter G200 device (model no. G200) manufactured by Agilent Technologies, Inc. (Chandler, AZ, USA) [27]. The measurement consisted of 30 indentation cycles, each applying a load of 50 mg for 15 s.

Following this step, the bulk alloys were pulverized using a high-energy vibratory mill. Our goal was to produce the material in the simplest possible and most cost-effective way. Therefore, the alloys were remelted five times to ensure sufficient elemental homogeneity of the feedstock, but without homogenization annealing. Some of the alloys were ductile and required a longer time to be pulverized in the vibratory mill, but this time did not exceed 20 min for any of the alloys. An additional density measurement was performed on this powder fraction using a helium pycnometer AccuPyc II 1345 Micromeritics. The results of both methods were, however, very similar, with differences only in the third decimal place.

The phase analysis of the powder samples was conducted using X-ray diffraction on a Philips X Pert Pro diffractometer (Malvern Panalytical, Almelo, The Netherlands). The measurements were performed in the 2θ range of 10° to 100°, with a step of 0.03° and a step time of 25 s. The X-ray wavelength of a copper anode is 1.5406 Å.

Hydrogen absorption and desorption measurements were conducted using a magnetic suspension balance (TA Instruments, New Castle, DE, USA) capable of operating at up to 50 MPa with an accuracy of 0.05%. The experiments were performed on all the samples using the following protocol:A known weight of powder alloy (~1 g per sample) was placed in the reaction chamber of the magnetic suspension balance. The system was sealed and evacuated to a vacuum of <5 Pa.The alloy was then activated by exposure to a low hydrogen pressure of ~100 kPa at room temperature for 1 h to reduce the oxides on the surfaces of the powder particles. To remove the absorbed hydrogen, the sample was then heated to 350 °C in a vacuum of ~2 Pa.Following activation, the first hydrogen absorption measurement was performed by reducing the sample temperature to 200 °C and by filling the reaction chamber with hydrogen to a pressure of 2 MPa. The increase in the sample’s weight due to hydrogen absorption was recorded for 1 h.After the absorption measurement, the reaction chamber was evacuated and the sample was heated to 370 °C, causing the desorption of hydrogen. The weight loss during this process was recorded.A second absorption cycle was conducted under the same conditions as the first.

The measured raw data were further recalculated for the amount of hydrogen absorbed per gram of material by using the typical routines for buoyancy corrections, as stated in [28].

## 3. Results and Discussion

The following text provides a detailed analysis of the obtained results. Table 1, Table 2 and Table 3 offer a comprehensive overview of the basic physical and chemical properties of the studied materials. These tables include key information such as density, composition, and mechanical properties. On the other hand, Table 4, Table 5 and Table 6 focus on presenting the results related to the thermodynamic properties of these materials, encompassing sorption capacities, and enthalpies. These data are crucial for a deeper understanding of the material properties and their potential industrial applications.

Before preparing the alloys, we employed well-established empirical rules to predict whether the target alloy compositions would fall into the supersaturated solid solution (medium/high entropy) region. To determine this, we calculated the theoretical values for mixing enthalpy, Δ*H_mix_*, applying Equation (1) [28].
(1)$${∆H}_{mix}=4\sum_{i=1,j\neq i}^{n}∆H_{<ij>}^{mix}c_{i}c_{j}$$
where *c_i_* and *c_j_* are the concentrations of elements *i* and *j*, and ${\Delta H}_{<ij>}^{mix}$ is the mixing enthalpy of elements *i* and *j* [29], and the differences in atomic sizes, *δ_r_*, are determined by applying Equation (2) [30].
(2)$$\delta_{r}=\sqrt{\sum_{i=1}^{n}\left\{c_{i}{\left(1-\frac{r_{i}}{\bar{r}}\right)}^{2}\right\}}\cdot 100\%$$where $\bar{r}$ is the average atomic radius, and *c_i_* is the atomic percentage of the *i*-th element with atomic radius *r_i_* [31,32,33]. All the calculations were performed by our prediction software developed on the MATLAB platform (https://ch.mathworks.com/products/matlab.html, accessed on 13 March 2023), incorporating all required referenced databases. The well-established empirical rules state that a single-phase solid solution (medium/high entropy alloys) is formed in the enthalpy interval −10 <${\;\Delta H}_{<ij>}^{mix}$ < 5 kJ/mol, as shown in [28].

The numerical outputs of our calculations are presented in Table 4, Table 5 and Table 6. Although the compositions of the alloys do not fall within the regions of highest probability for forming supersaturated solid solutions, previous studies have shown that these alloys can form single phases [34].

Figure 1, Figure 2 and Figure 3 show the graphical representation of the empirical rules, with the plotted Al-Ti-Nb-Zr, Al-Ti-Nb-V, and Al-Ti-Nb-Hf alloys, respectively. Since the alloys do not fall into the region of stability for highly supersaturated solid solutions, according to their thermodynamic parameters, but can nevertheless form them, it can only be stated that they deviate from the most widely used empirical predictive theory for medium- and high-entropy alloys.

Table 1, Table 2 and Table 3 provide data about the fundamental material characteristics of the alloys. The first column lists the targeted composition of each alloy, while the second column presents the actual chemical elemental composition determined by energy-dispersive X-ray spectroscopy (EDS) after alloy production. The actual compositions of the alloys differ from the target compositions by no more than 3 at.%.

The third column provides the density values of the powder alloys, determined by the pycnometric method. The listed values represent the average of ten measurements, supplemented also with standard error estimation.

The density of the alloys in the Al-Ti-Nb-Zr group ranged from 5.31 to 6.11 g/cm^3^ (Table 1). The lowest density was observed for the Al_30_Ti_35_Nb_15_Zr_20_ sample, with a value of 5.31 g/cm^3^. The second group of alloys, consisting of Al-Ti-Nb-V elements, displayed densities ranging from 5.1 to 6.74 g/cm^3^ (Table 2). Here, the Al_25_Ti_35_Nb_15_V_25_ alloy exhibited the lowest density of 5.1 g/cm^3^ among all the investigated alloys, making it a promising candidate for applications in the transportation sector. The last group of investigated alloys, with compositions of Al-Ti-Nb-Hf elements, displayed the highest density values among the produced alloys, ranging from 6.23 to 7.67 g/cm^3^ (Table 3). The Al_30_Ti_35_Nb_15_Hf_20_ alloy demonstrated the lowest density of 6.23 g/cm^3^ in this group (Table 3).

In contrast to other well-studied medium-entropy alloys, our investigated alloys, particularly those from the Al-Ti-Nb-V group, belong to lightened alloys. For example, HEAs based on iron, chromium, manganese, nickel, and aluminum have densities of around 7.5 g·cm^−3^; alloys based on chromium, cobalt, copper, nickel, and iron reach a value of 7.2 g·cm^−3^; and alloys based on iron, aluminum, chromium, nickel, and copper have a value of 7.3 g·cm^−3^ [35,36].

Columns four and five present the microhardness and nanohardness values of the alloys, respectively. The microhardness of the Al-Ti-Nb-Zr alloys range from 353 to 747.8 (Table 1), while those with Al-Ti-Nb-V composition exhibit a range of 320.7 to 500.3 (Table 2). The hardness of the Al-Ti-Nb-Hf composition falls within the range of 347.9 to 801.7 (Table 3). Notably, these alloy systems demonstrate significant hardness variations with minor changes in their chemical composition.

Among the Al-Ti-Nb-Zr alloys, the Al_15_Ti_38_Nb_23_Zr_24_ sample exhibited the lowest nanohardness value, reaching 4.4 GPa (Table 1). In the Al-Ti-Nb-V group, the Al_15_Ti_40_Nb_25_V_20_ alloy had the lowest value of 4.3 GPa (Table 2), and within the Al-Ti-Nb-Hf group, the alloy Al_15_Ti_30_Nb_25_Hf_30_ sample had the lowest nanohardness of 4.2 ± 0.26 GPa (Table 3).

Modern nanohardness measurements also provide, in addition to hardness measurements, valuable information about a material’s elastic modulus E, representing the limit of its elastic deformation as a function of external forces [37,38]. Our measurements revealed that the *E* values of the Al-Ti-Nb-Zr alloys ranged from 90 to 160 GPa (Table 1); for the Al-Ti-Nb-V alloys, they ranged from 130 to 160 GPa (Table 2); and for the Al-Ti-Nb-Hf alloys, from 110 to 180 GPa (Table 3). These findings indicate that all these alloys possess a significant degree of resistance to elastic deformation (stiffness). For comparison, steels exhibit elastic moduli in the range of 190–210 GPa, titanium alloys in a range from 100 to 120 GPa, and aluminum alloys fall within a 70–80 GPa range [39,40,41].

Figure 4, Figure 5 and Figure 6 depict the X-ray powder diffraction data of the as-prepared alloys (black curves) and after hydrogen exposure (red curves). While most of the as-prepared alloys exhibit a single BCC phase, some exhibit a split main peak and/or an adjacent satellite peak, indicating the presence of additional phases, such as secondary BCC and/or FCC (Face-Centered Cubic).

By a visual comparison, the hydrogen-exposed alloys appear very similar, sometimes identical, to the patterns of the as-prepared state. This could be attributed to either the samples not absorbing any hydrogen, or that they did absorb hydrogen (as will be shown later), but spontaneously released it at ambient conditions. Spontaneous hydrogen desorption is advantageous for storage tank applications, since the amount of absorbed hydrogen can be controlled by adjusting the ambient hydrogen gas pressure, without the need for additional heating of the alloy. However, the XRD patterns of some of the alloys after hydrogenation exhibit a significant change, with the Bragg peaks shifted to the left (i.e., an increase in the unit cell in the crystal lattice). In other samples, completely new Bragg peaks appear, indicating the formation of new hydride phases with different lattice structures.

As is known, the low intensity and broadening (FWHM) of diffraction peaks are manifestations of high internal stresses and small crystallite size. In our opinion, the primary reason for the defective structure, exhibiting a small size of coherently diffracting crystallites, is not milling (as this is rather a breaking of a massive buttons), but rapid heat removal from the melt to the cooled Cu substrate. As can be seen from the XRD patterns, hydrogenation in most cases did not cause a large change in the XRD profile. The small-angle contribution most probably comes from the plexiglass sample holder.

Columns two through five of Table 4, Table 5 and Table 6 provide the thermodynamic parameters for all the prepared alloys, including their mixing enthalpy, mixed atomic radius ratio, mixing entropy, and valence electron concentration.

Column six of Table 4, Table 5 and Table 6 lists the maximum hydrogen absorption capacity of each alloy. These values are expressed in terms of both the weight increase (wt.%) and the number of hydrogen atoms per one metallic atom (H/M ratio, column seven), calculated by using the following Equation (3) [42].
(3)$$H/M=\frac{c_{wt.\;\%}\cdot M_{host}}{M_{H}\cdot 100-c_{wt.\%}\cdot M_{H}}$$
where *c*_*wt.*%_ is the hydrogen capacity in wt.%, *M_H_* is the molar mass of hydrogen, and *M_host_* is the molar mass of the host material or alloy.

The column labeled “Residual” shows the amount of hydrogen that remains in the alloy after desorption (heating to 370 °C in a vacuum for 2 h). The amount of desorbed hydrogen from the alloy is given in column eight, “Desorption”. The value in column no. nine, “Cycle efficiency”, indicates the efficiency of the absorption/desorption cycle, which is expressed as the ratio of the desorption and absorption values multiplied by 100. This value quantifies how well a specific alloy can effectively desorb hydrogen from its volume under the specified testing conditions.

The following can be concluded from the comparison of these values:The Al_15_Ti_38_Nb_23_Zr_24_ alloy is the most hydrogen-absorbing alloy in the Al-Ti-Nb-Zr system, with a maximum absorption of 1.61 wt.% (H/M = 1.05). However, after annealing at 370 °C in a vacuum, it still retains up to 0.62 wt.% (H/M = 1.05) of chemically bounded hydrogen, meaning that the amount of reversibly released hydrogen is only 0.99 wt.% (H/M = 0.65). From this perspective, the Al_15_Ti_40_Nb_30_Zr_15_ alloy is more advantageous, with a maximum reversible release of 1.03 wt.% of hydrogen (H/M = 0.68).In the case of the Al-Ti-Nb-V system, the Al_15_Ti_40_Nb_25_V_20_ alloy exhibits a maximum capacity of reversibly storable hydrogen of 1.02 wt.% (H/M = 0.57).In the Al-Ti-Nb-Hf system, the best is the Al_15_Ti_40_Nb_30_Hf_20_ alloy, which has a maximum capacity of reversibly storable hydrogen of 0.82 wt.% (H/M = 0.69).Considering the densities of the Al_15_Ti_40_Nb_30_Zr_15_, Al_15_Ti_40_Nb_25_V_20_, and Al_15_Ti_40_Nb_30_Hf_20_ alloys (6.11 g·cm^−3^, 5.646 g·cm^−3^, and 7.041 g·cm^−3^, respectively), the most efficient alloy appears to be Al_15_Ti_40_Nb_30_Zr_15_. It is 8.2% heavier than the vanadium alloy, but it has a significantly higher amount of reversibly storable hydrogen—H/M = 0.68, compared to H/M = 0.57 for the Al_15_Ti_40_Nb_25_V_20_. The best Hf-containing sample has a very similar amount of reversibly storable hydrogen as the Zr-containing sample, but it is 15.2% heavier.

For a comparison with our results, the most widely used commercial alloy for reversible hydrogen storage today is LaNi_5_, which can store ~1.2 wt.% of hydrogen at a hydrogen pressure of 2 MPa and temperature of 110 °C (which are approximately the conditions of our experiment) [43].

This corresponds to an H/M = 1.15. The alloy Al_15_Ti_40_Nb_30_Zr_15_ has a reversible hydrogen storage capacity that is 40% lower than that of LaNi_5_, but it is 23% lighter and does not contain rare earth elements.

The absorption/desorption measurements for the Al_15_Ti_40_Nb_30_Zr_15_ sample are shown in Figure 7, where the upper-left quadrant shows in blue the measured change in the sample weight, together with the temperature profile during the absorption and desorption cycles (red curve). The normalized absorption corrected for the hydrogen buoyancy is shown in the upper-right quadrant. Here the yellow-marked areas show the state of the system when it is not in equilibrium (for example, during heating and/or during hydrogen charging of the reaction chamber). The absorption of hydrogen, in the ratio H/M, is shown in the lower-left quadrant. The fourth quadrant compares the absorption kinetics of the first and second cycles. It is necessary to pay attention to a certain phenomenon that was observed in all our samples. The fully saturated state in these alloys is achievable within 3 min under conditions of 2 MPa of H_2_ and a temperature of 200 °C.

Since this study presents results for a large number of samples (up to 30 alloys, on which different types of measurements were performed), it is possible to verify the correlation between the material and its thermodynamic parameters and its ability to absorb or desorb hydrogen. Figure 8, Figure 9 and Figure 10 show the relationship between the density (*ρ*), mixing enthalpy (Δ*H_mix_*), valence electron concentration (*VEC*), microhardness (*HV*_0.3_), nanohardness (*H_in_*), and elastic modulus (*E*) and the maximum hydrogen absorption of the alloy, and the amount of desorbed hydrogen from the alloys of all the prepared systems. The gray part indicates the observed trend in the dependencies between individual quantities.

Based on this comparison, several trends can be seen. For example, it seems that both the maximum amount of absorbed hydrogen and the amount of desorbable hydrogen from an alloy (annealed for two hours at 370 °C in vacuum) increases with increasing Δ*H_mix_* and *VEC* values. However, the correlation between these sorption properties and microhardness *HV*_0.3_ is very clear. As the microhardness increases, the amount of absorbed and desorbed hydrogen from an alloy decreases. Similarly, but less pronounced, the trend is also observed for *H_in_* and *E*. To the best of our knowledge, these dependencies have not yet been presented in the scientific literature.

## 4. Conclusions

In this scientific paper, we reported the design, preparation, and characterization of 30 completely new (unpublished) medium-entropy alloys of three systems: Al-Ti-Nb-Zr, Al-Ti-Nb-V, and Al-Ti-Nb-Hf. The motivation was to verify the hydrogen absorption capacity of lightened alloys having a density below 7.7 g·cm^−3^. We performed a comprehensive material characterization of all the alloys, including chemical, phase analysis, density hardness, nanohardness, and elastic modulus measurements. All the alloys were also characterized in terms of their hydrogen absorption and desorption abilities. The main conclusions of our work can be summarized in the following points:Despite the fact that empirical rules suggested that the alloys would fall outside the stability range of single-phase solid solutions, most of the alloys are single-phase with a simple BCC structure.The hardness of the Al-Ti-Nb-X alloys (X = Zr, V, or Hf) ranges from 320 to 800 HV_0.3_ units.The most promising alloy is Al_15_Ti_40_Nb_30_Zr_15_. It has the highest-reversible hydrogen storage capacity of all the alloys, at 1.03 wt.% of hydrogen, corresponding to and H/M = 0.68. This is 40% lower than the reversible hydrogen storage capacity of LaNi_5_, but Al_15_Ti_40_Nb_30_Zr_15_ is 23% lighter and does not contain rare earth elements.Since this work summarizes the results for a relatively large number of alloys produced in the same way, it was possible to verify the correlation between the material, its thermodynamic characteristics, and the sorption properties of these alloys. Among all the verified dependencies, the most pronounced trend shows the influence of hardness (micro and nano) on the absorption or desorption of hydrogen into or from the alloys. With the increasing hardness of the alloys, the amount of absorbed or desorbed hydrogen decreases. This observation is fundamentally understandable. Hardness is a macroscopical property of the strength of interatomic bonds, so it can be assumed that the stronger the bonds, the more difficult it is for hydrogen to penetrate into the volume of the alloy and create a hydride phase there. To our knowledge, this phenomenon has not been reported before, and this article provides experimental evidence for it.

## Figures and Tables

**Figure 1 materials-17-02897-f001:**
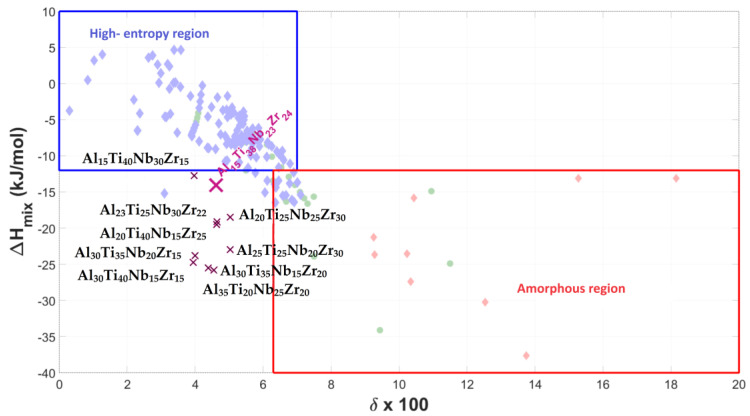
Graphical representation of phase stability regions for predicting formation of supersaturated solid solutions (medium/high entropy) in Al-Ti-Nb-Zr materials, with representative alloy compositions plotted.

**Figure 2 materials-17-02897-f002:**
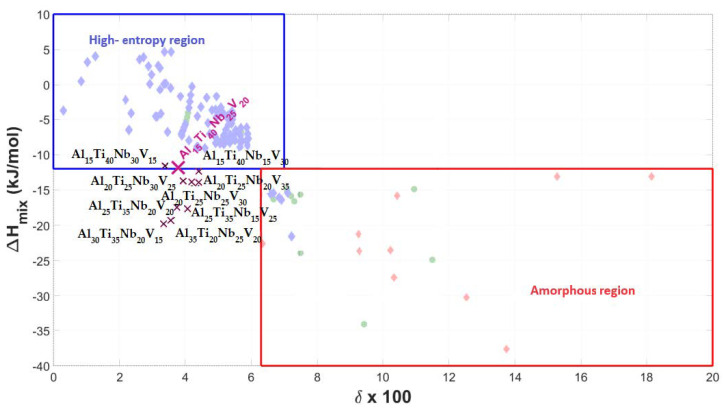
Graphical representation of phase stability regions for predicting formation of supersaturated solid solutions (medium–high entropy) in Al-Ti-Nb-V materials, with representative alloy compositions plotted.

**Figure 3 materials-17-02897-f003:**
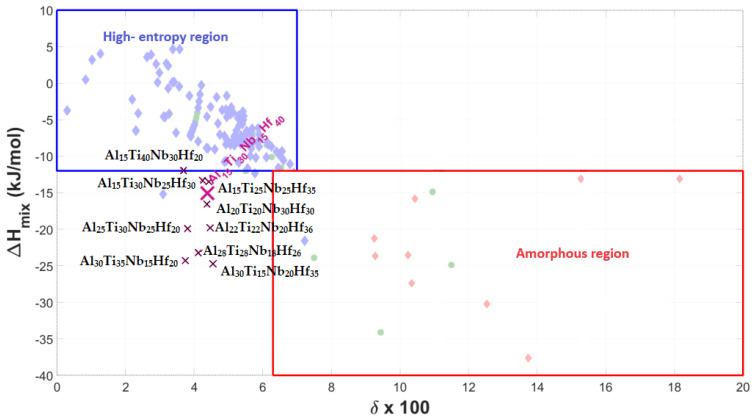
Graphical representation of phase stability regions for predicting formation of supersaturated solid solutions (medium–high entropy) in Al-Ti-Nb-Hf materials, with representative alloy compositions plotted.

**Figure 4 materials-17-02897-f004:**
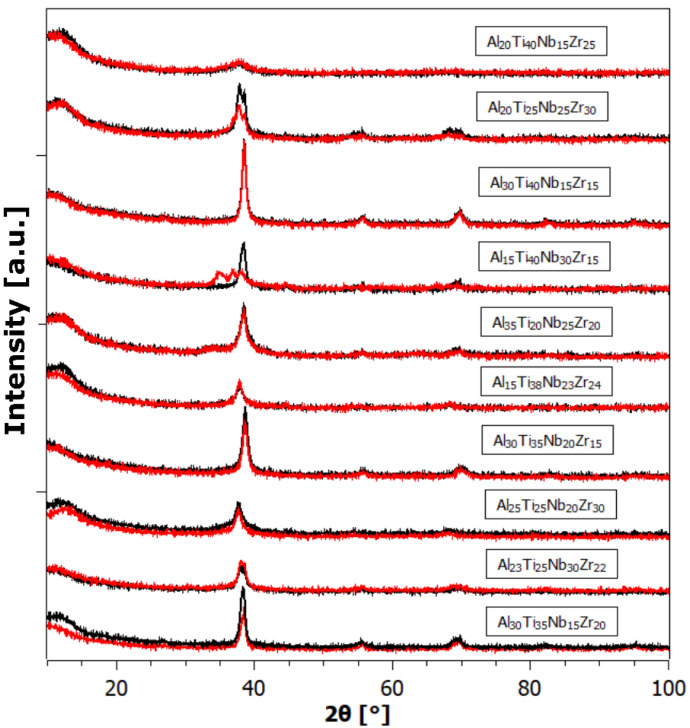
The XRD powder pattern of samples from the Al-Ti-Nb-Zr system in the as-prepared state (black) and after hydrogenation (red).

**Figure 5 materials-17-02897-f005:**
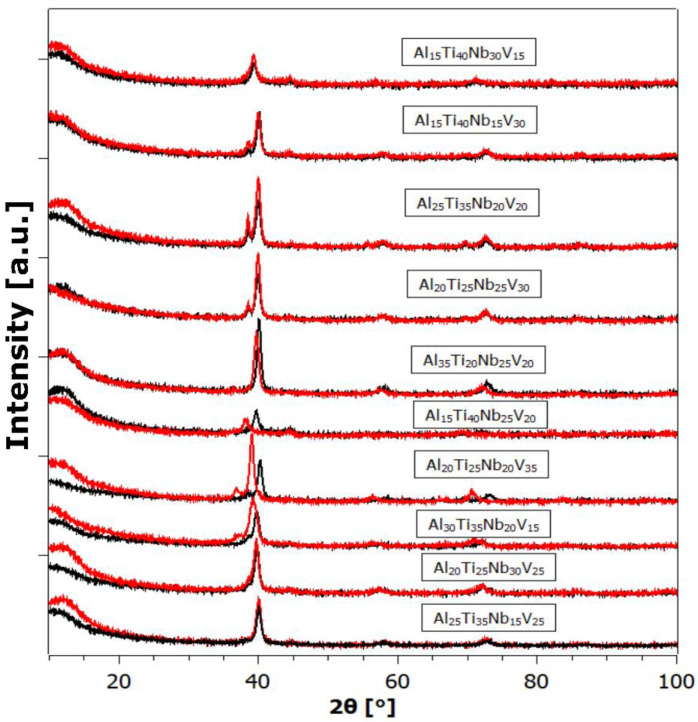
The XRD powder pattern of samples from the Al-Ti-Nb-V system in the as-prepared state (black) and after hydrogenation (red).

**Figure 6 materials-17-02897-f006:**
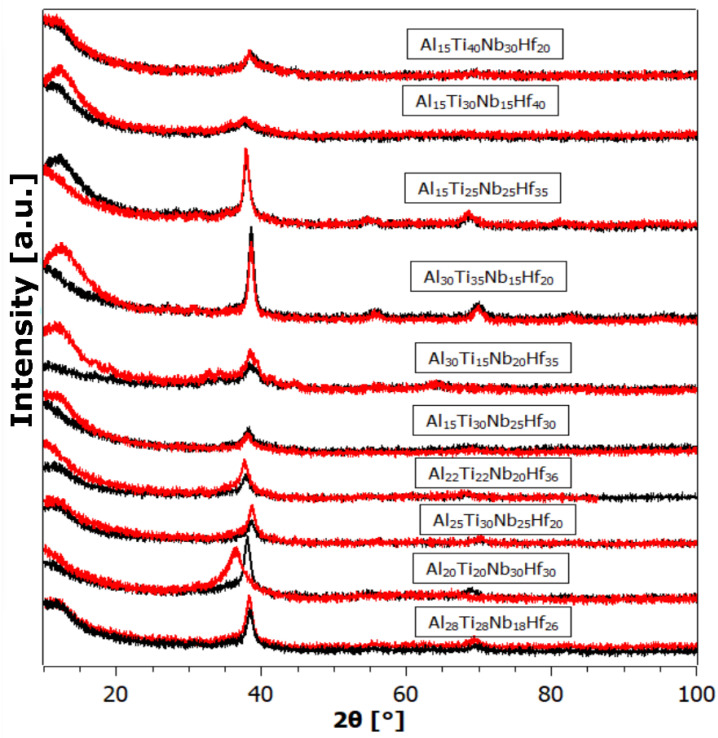
The XRD powder pattern of samples from the Al-Ti-Nb-Hf system in the as-prepared state (black) and after hydrogenation (red).

**Figure 7 materials-17-02897-f007:**
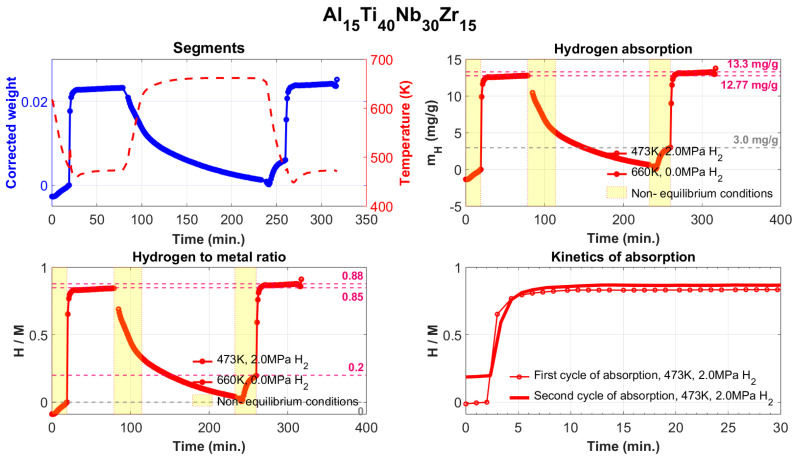
The course of the absorption/desorption measurements of the Al_15_Ti_40_Nb_30_Zr_15_ alloy.

**Figure 8 materials-17-02897-f008:**
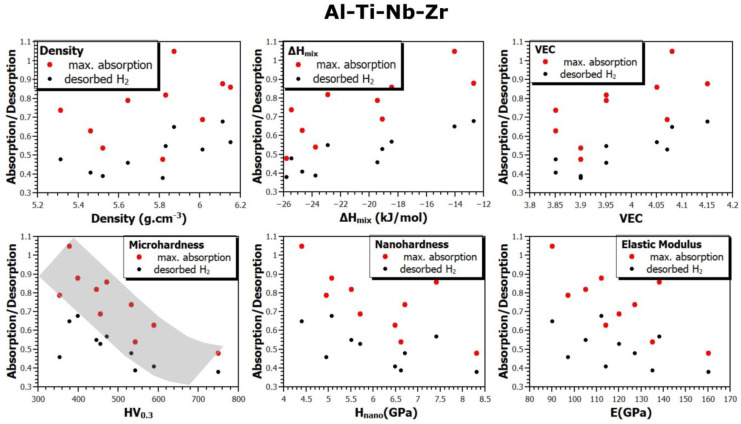
The relationship between material and thermodynamic parameters and hydrogen absorption and desorption in the Al-Ti-Nb-Zr system.

**Figure 9 materials-17-02897-f009:**
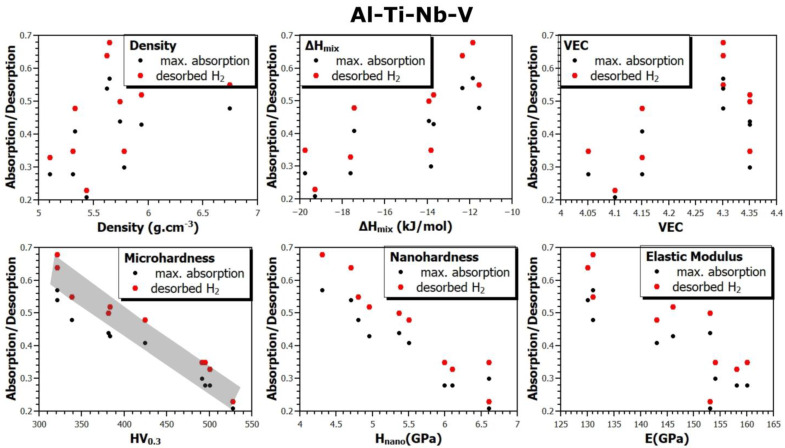
The relationship between material and thermodynamic parameters and hydrogen absorption and desorption in the Al-Ti-Nb-V system.

**Figure 10 materials-17-02897-f010:**
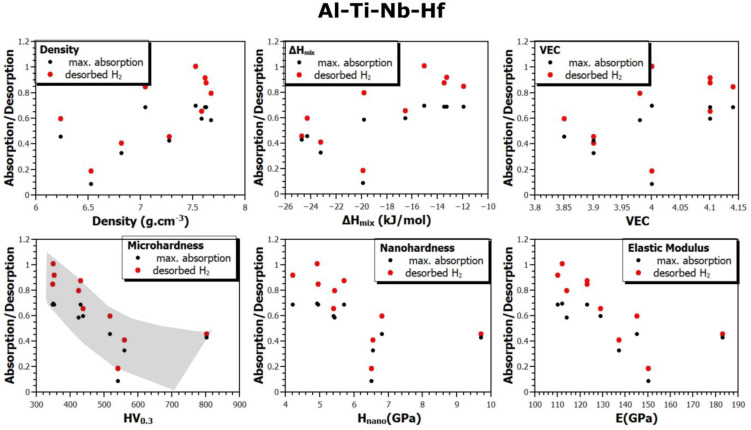
The relationship between material and thermodynamic parameters and hydrogen absorption and desorption in the Al-Ti-Nb-Hf system.

**Table 1 materials-17-02897-t001:** Chemical composition, density, and mechanical properties of the Al-Ti-Nb-Zr alloys.

Alloy	EDX [at.%]	Density [g·cm^−3^]	Microhardness HV_0.3_	Nanohardness [GPa]	Modulus of Elasticity [GPa]
Al_30_Ti_35_Nb_15_Zr_20_	Al_30_Ti_34_Nb_14_Zr_22_	5.312 ± 0.009	532 ± 93	6.7 ± 0.7	127 ± 10
Al_23_Ti_25_Nb_30_Zr_22_	Al_23_Ti_27_Nb_28_Zr_22_	6.011 ± 0.006	454 ± 44	5.7 ± 0.2	120 ± 3.7
Al_30_Ti_35_Nb_20_Zr_15_	Al_31_Ti_36_Nb_20_Zr_13_	5.52 ± 0.14	542 ± 21	6.61 ± 0.02	135 ± 5
Al_25_Ti_25_Nb_20_Zr_30_	Al_24_Ti_26_Nb_21_Zr_30_	5.83 ± 0.02	444 ± 7	5.5 ± 0.12	105 ± 2
Al_15_Ti_38_Nb_23_Zr_24_	Al_16_Ti_38_Nb_23_Zr_23_	5.87 ± 0.012	377 ± 9	4.4 ± 0.12	90 ± 3
Al_35_Ti_20_Nb_25_Zr_20_	Al_33_Ti_23_Nb_24_Zr_20_	5.815 ± 0.008	748 ± 31	8.3 ± 0.77	160 ± 8
Al_30_Ti_40_Nb_15_Zr_15_	Al_28_Ti_38_Nb_16_Zr_18_	5.46 ± 0.017	588 ± 19	6.48 ± 0.06	114 ± 2
Al_20_Ti_25_Nb_25_Zr_30_	Al_21_Ti_26_Nb_26_Zr_27_	6.15 ± 0.019	470 ± 141	7.4 ± 0.11	138 ± 5
Al_20_Ti_40_Nb_15_Zr_25_	Al_19_Ti_41_Nb_15_Zr_25_	5.643 ± 0.004	353 ± 11	4.94 ± 0.08	97 ± 4
Al_15_Ti_40_Nb_30_Zr_15_	Al_14_Ti_39_Nb_29_Zr_18_	6.11 ± 0.02	398 ± 21	5.07 ± 0.12	112 ± 3

**Table 2 materials-17-02897-t002:** Chemical composition, density, and mechanical properties of the Al-Ti-Nb-V alloys.

Alloy	EDX [at.%]	Density [g·cm^−3^]	Microhardness HV_0.3_	Nanohardness [GPa]	Modulus of Elasticity [GPa]
Al_25_Ti_35_Nb_15_V_25_	Al_26_Ti_36_Nb_13_V_25_	5.101 ± 0.001	500 ± 10	6.1 ± 0.12	158 ± 5
Al_20_Ti_25_Nb_30_V_25_	Al_18_Ti_26_Nb_30_V_27_	5.934 ± 0.007	383 ± 42	4.95 ± 0.06	146 ± 3
Al_30_Ti_35_Nb_20_V_15_	Al_27_Ti_35_Nb_22_V_16_	5.31 ± 0.013	495 ± 14	5.99 ± 0.09	160 ± 3
Al_20_Ti_25_Nb_20_V_35_	Al_18_Ti_24_Nb_22_V_35_	5.74 ± 0.012	381 ± 34	5.36 ± 0.07	153 ± 3
Al_15_Ti_40_Nb_25_V_20_	Al_16_Ti_38_Nb_23_V_22_	5.646 ± 0.006	321 ± 8	4.3 ± 0.18	131 ± 3
Al_35_Ti_20_Nb_25_V_20_	Al_34_Ti_19_Nb_25_V_22_	5.432 ± 0.009	527 ± 14	6.6 ± 0.16	153 ± 5
Al_20_Ti_25_Nb_25_V_30_	Al_18_Ti_25_Nb_27_V_30_	5.78 ± 0.01	491 ± 21	6.6 ± 0.13	154 ± 2
Al_25_Ti_35_Nb_20_V_20_	Al_25_Ti_36_Nb_19_V_20_	5.330 ± 0.007	424 ± 15	5.5 ± 0.11	143 ± 3
Al_15_Ti_40_Nb_15_V_30_	Al_13_Ti_40_Nb_15_V_31_	5.620 ± 0.009	321 ± 7	4.7 ± 0.1	130 ± 2
Al_15_Ti_40_Nb_30_V_15_	Al_14_Ti_43_Nb_29_ V_15_	6.74 ± 0.012	338 ± 18	4.8 ± 0.09	131 ± 3

**Table 3 materials-17-02897-t003:** Chemical composition, density, and mechanical properties of the Al-Ti-Nb-Hf alloys.

Alloy	EDX [at.%]	Density [g·cm^−3^]	Microhardness HV_0.3_	Nanohardness [GPa]	Modulus of Elasticity [GPa]
Al_28_Ti_28_Nb_18_Hf_26_	Al_28_Ti_28_Nb_18_Hf_26_	6.813 ± 0.007	559 ± 15	6.54 ± 0.09	137 ± 6
Al_20_Ti_20_Nb_30_Hf_30_	Al_21_Ti_21_Nb_30_Hf_29_	7.58 ± 0.01	437 ± 7	5.4 ± 0.19	129 ± 1
Al_25_Ti_30_Nb_25_Hf_20_	Al_26_Ti_31_Nb_24_Hf_19_	6.522 ± 0.008	540 ± 41	6.5 ± 0.17	150 ± 3
Al_22_Ti_22_Nb_20_Hf_36_	Al_22_Ti_22_Nb_20_Hf_36_	7.67 ± 0.02	424 ± 45	5.42 ± 0.04	114 ± 4
Al_15_Ti_30_Nb_25_Hf_30_	Al_18_Ti_32_Nb_23_Hf_28_	7.612 ± 0.007	351 ± 5	4.2 ± 0.26	110 ± 4
Al_30_Ti_15_Nb_20_Hf_35_	Al_31_Ti_16_Nb_21_Hf_32_	7.27 ± 0.012	801 ± 57	9.7 ± 1.7	183 ± 23
Al_30_Ti_35_Nb_15_Hf_20_	Al_30_Ti_33_Nb_17_Hf_20_	6.234 ± 0.008	516 ± 8	6.8 ± 0.1	145 ± 4
Al_15_Ti_25_Nb_25_Hf_35_	Al_15_Ti_25_Nb_27_Hf_34_	7.622 ± 0.006	430 ± 9	5.7 ± 0.9	123 ± 14
Al_15_Ti_30_Nb_15_Hf_40_	Al_17_Ti_31_Nb_15_Hf_37_	7.523 ± 0.018	349 ± 6	4.91 ± 0.08	112 ± 3
Al_15_Ti_40_Nb_30_Hf_20_	Al_14_Ti_36_Nb_32_Hf_19_	7.041 ± 0.008	347 ± 5	4.95 ± 0.09	123 ± 2

**Table 4 materials-17-02897-t004:** Thermodynamic parameters of Al-Ti-Ni-Zr alloys and hydrogen absorption test results.

Alloy	ΔH_mix_ [kJ/mol]	δ × 100	ΔS_mix_	VEC	Absorption H [wt.%]/[H/M]	Residual H [wt.%]/[H/M]	Desorption H [wt.%]/[H/M]	Cycle Efficiency [%]
Al_30_Ti_35_Nb_15_Zr_20_	−25.50	4.39	11.10	3.85	1.28/0.74	0.46/0.26	0.82/0.48	64.86
Al_23_Ti_25_Nb_30_Zr_22_	−19.11	4.64	11.46	4.07	1.06/0.69	0.25/0.16	0.81/0.53	76.91
Al_30_Ti_35_Nb_20_Zr_15_	−23.80	3.99	11.10	3.90	0.95/0.54	0.27/0.15	0.71/0.39	72.22
Al_25_Ti_25_Nb_20_Zr_30_	−22.94	5.03	11.44	3.95	1.23/0.82	0.41/0.27	0.82/0.55	67.07
Al_15_Ti_38_Nb_23_Zr_24_	−14.08	4.61	11.08	4.08	1.61/1.05	0.62/0.40	0.99/0.65	61.90
Al_35_Ti_20_Nb_25_Zr_20_	−25.82	4.54	11.29	3.90	0.79/0.48	0.18/0.10	0.61/0.38	79.16
Al_30_Ti_40_Nb_15_Zr_15_	−24.72	3.95	10.78	3.85	1.10/0.63	0.37/0.22	0.73/0.41	65.08
Al_20_Ti_25_Nb_25_Zr_30_	−18.46	5.04	11.44	4.05	1.28/0.86	0.43/0.29	0.85/0.57	66.28
Al_20_Ti_40_Nb_15_Zr_25_	−19.48	4.62	10.97	3.95	1.28/0.79	0.54/0.33	0.74/0.46	58.23
Al_15_Ti_40_Nb_30_Zr_15_	−12.72	3.96	10.78	4.15	1.33/0.88	0.30/0.20	1.03/0.68	77.27

**Table 5 materials-17-02897-t005:** Thermodynamic parameters of Al-Ti-Ni-V alloys and hydrogen absorption test results.

Alloy	ΔH_mix_ [kJ/mol]	δ × 100	ΔS_mix_	VEC	Absorption H [wt.%]/H/M	Residual H [wt.%]/H/M	Desorption H [wt.%]/ H/M	Cycle Efficiency [%]
Al_25_Ti_35_Nb_15_V_25_	−17.63	4.08	11.18	4.15	0.68/0.33	0.1/0.05	0.58/0.28	84.85
Al_20_Ti_25_Nb_30_V_25_	−13.72	3.94	11.44	4.35	0.88/0.52	0.15/0.09	0.73/0.43	82.69
Al_30_Ti_35_Nb_20_V_15_	−19.78	3.35	11.10	4.05	0.67/0.35	0.13/0.07	0.54/0.28	80.00
Al_20_Ti_25_Nb_20_V_35_	−13.94	4.41	11.29	4.35	0.91/0.50	0.12/0.06	0.79/0.44	88.00
Al_15_Ti_40_Nb_25_V_20_	−11.86	3.79	10.97	4.30	1.23/0.68	0.21/0.11	1.02/0.57	83.82
Al_35_Ti_20_Nb_25_V_20_	−19.30	3.57	11.28	4.10	0.44/0.23	0.04/0.02	0.4/0.21	91.3
Al_20_Ti_25_Nb_25_V_30_	−13.84	4.20	11.44	4.35	0.61/0.35	0.09/0.05	0.52/0.3	85.71
Al_25_Ti_35_Nb_20_V_20_	−17.46	3.75	11.29	4.15	0.92/0.48	0.13/0.07	0.79/0.41	85.42
Al_15_Ti_40_Nb_15_V_30_	−12.36	4.40	10.78	4.30	1.2/0.64	0.19/0.1	1.01/0.54	84.38
Al_15_Ti_40_Nb_30_V_15_	−11.58	3.39	10.78	4.30	0.95/0.55	0.13/0.07	0.82/0.48	27.27

**Table 6 materials-17-02897-t006:** Thermodynamic parameters of Al-Ti-Ni-Hf alloys and hydrogen absorption test results.

Alloy	ΔH_mix_ [kJ/mol]	δ × 100	ΔS_mix_	VEC	Absorption H [wt.%]/H/M	Residual H [wt.%]/H/M	Desorption H [wt.%]/ H/M	Cycle Efficiency [%]
Al_28_Ti_28_Nb_18_Hf_26_	−23.24	4.11	11.41	3.90	0.63/0.41	0.10/0.08	0.53/0.33	80.34
Al_20_Ti_20_Nb_30_Hf_30_	−16.56	4.37	11.36	4.10	0.69/0.66	0.06/0.06	0.63/0.60	90.91
Al_25_Ti_30_Nb_25_Hf_20_	−19.90	3.81	11.44	4.00	0.24/0.19	0.21/0.10	0.03/0.09	47.37
Al_22_Ti_22_Nb_20_Hf_36_	−19.83	4.47	11.27	3.98	0.80/0.80	0.22/0.21	0.58/0.59	73.75
Al_15_Ti_30_Nb_25_Hf_30_	−13.32	4.24	11.25	4.10	1.01/0.92	0.25/0.23	0.76/0.69	75.00
Al_30_Ti_15_Nb_20_Hf_35_	−24.74	4.55	11.10	3.90	0.50/0.46	0.03/0.03	0.47/0.43	93.48
Al_30_Ti_35_Nb_15_Hf_20_	−24.30	3.75	11.10	3.85	0.80/0.60	0.19/0.14	0.61/0.46	76.67
Al_15_Ti_25_Nb_25_Hf_35_	−13.49	4.42	11.18	4.10	0.86/0.88	0.19/0.19	0.67/0.69	78.41
Al_15_Ti_30_Nb_15_Hf_40_	−15.06	4.38	10.78	4.00	1.02/1.01	0.32/0.31	0.70/0.70	69.30
Al_15_Ti_40_Nb_30_Hf_20_	−11.97	3.68	10.97	4.14	1.01/0.85	0.19/0.16	0.82/0.69	81.18

## Data Availability

The raw data supporting the conclusions of this article will be made available by the authors on request.

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
