# Peer review of "New-Generation Materials for Hydrogen Storage in Medium-Entropy Alloys"

_materials, 2024, doi:10.3390/ma17122897_

Round 1

Reviewer 1 Report

Comments and Suggestions for Authors

In this paper the Authors have synthesized and characterized 30 medium entropy alloys, and investigated their hardness and hydrogenation properties. The samples were fabricated by arc-melting and pulverized to powders with high energy ball-milling prior to the hydrogen absorption and desorption experiments. The hardness was measured on as-cast samples, but except from storage capacity and XRD, no further characterization of the milled powders is provided.

The mechanical hardness of an alloy varies depending on the specific alloy composition and its processing history. Different elements added to the base metal can significantly impact hardness, and processes like heat treatment, cold working, and aging can significantly alter the microstructure of an alloy, and consequently, its hardness. The Authors have performed hardness measurement on as-cast samples, while hydrogenation has been performed after high energy ball milling. This process is a severe plastic deformation-based method abundantly reported to induce significant microstructural refinement which is known to alter the hydrogen ab/desorption properties of alloys. Therefore, the experimental design proposed herein is invalid in the sense that the hardness measured on as-cast samples is different from the actual hardness of ball-milled samples. It is hence impossible to correlate hardness increase to the decrease in hydrogen storage capacity. Furthermore, Authors have performed EDX but do not show any microstructural images that could help to understand the hydrogenation behavior.

Medium/High entropy alloys are known to exhibit sloping equilibrium plateau pressures (measured by pressure-composition-temperature experiments), and such sloping tends to be more significant as the number of constituents increases and in the absence of solution heat treatment after arc-melting. Given the sloping in the equilibrium plateau pressure, applying a fixed pressure for hydrogen absorption experiment will result in a decrease of the storage capacity as the sloping increases. Therefore, sloping (and hence compositional gradient/inhomogeneities) are more likely to result in a decrease of the hydrogen storage capacity rather than the hardness. Microstructural observations, including EDS mapping would have provided sufficient evidences but Authors only shared EDS data rather than maps and micrographs. For these reasons I believe this study is inconclusive (at least item 4 of the conclusions), and do not recommend publication in Materials. Please find additional comments attached below:

1.The abstract is informative and concise, and gathers all main findings of the paper in a straightforward manner. The introduction is clear, well written, and gives a good overview of the European strategy on hydrogen, as well as hydrogen production, utilization and current applications on the continent. However, the introduction critically lacks of scientific background. The pioneering work from Sahlberg’s group indeed provided a systematic study on the interplay between structure, microstructure, composition and the resulting hydrogenation properties of “classic” high entropy alloys, however the field of metal hydrides and hydrogen solid-state storage is not limited to this unique impactful study. The state-of-the-art on other types of high entropy alloys, medium entropy alloys, room temperature Abx-type intermetallic compounds, and other more complex hydrides (Mg, Li or B-based) etc. is missing. The introduction needs to be significantly revised by extending the literature review and perhaps reducing the background on EU’s strategy, to match the format of a scientific paper and not that of a grant application.

2.The experimental section is clear and well written as well. More details regarding the origin of the starting elements is however required. Besides. After arc-melting, were the samples heat treated to improve homogeneity? Furthermore, why carrying out ball-milling for as long as 20 min for pulverization?> As you know, this pulverization process will introduce significant stress/strain in the obtained powders, which in turn will affect both the hardness and the hydrogenation properties. were any measures taken to avoid this phenomenon?

3.Why not performing the activation procedure on coarse granule rather than fine powder? The pulverization process will provide a significantly larger total surface area that may oxidize faster, which in turn might make the activation process more difficult (requiring longer time or significantly higher temperatures). Can the Authors comment on that?

4.Can the Authors comment on the stability of the hydride phases? As per the experimental procedure, elevated temperatures are applied to hydrogenate and dehydrogenate the samples, which is impractical for large-scale industrial applications (although room temperature reversibility is claimed). Have the Authors considered acquiring PCT plots to calculate enthalpy and entropy of hydride formation and discuss hydride stability?

5.Fiugures 1, 2 and 3 need to be revised for a clear visualization: the text seems a bit too small and colors for the data are too light (hence difficult for readers with visual impairment and difficult to distinguished if the paper is printed without colors). The figures could bring a lot of insights in the solid-solution forming ability of the alloys but are barely described and unfortunately not discussed.

6.Please check the consistency in the terminology used when describing the density of the synthesized samples. Density, gravimetry, and event gravity(?) are used. Furthermore, I find it unlikely that the Medium entropy alloys developed here can be considered light weight materials and hence usable in the mobility sector, these alloys are rather heavy, and although arguably lighter than some others, the low hydrogen gravimetric density (~1.8wt % H2) will require larger volumes of alloys to store the required amount of hydrogen to propel a vehicle (~3kg H2). This would roughly yield a storage tank as heavy as 166 kg, to which the fuel cell weight and that of all other car components have to be added. For this reason it is difficult to claim these alloys to be “light weight alloys suitable for  the mobility sector. Please comment on that and revise your manuscript accordingly.

7.The XRD patterns (Figure 4 to 6) are insufficiently discussed and many interrogations remain. First of all, why are the diffraction peaks so broad? Does it originate from the lattice distortion / microstructural refinement induced by the harsh 20min high energy ball milling process or from the chemical inhomogeneity due to the absence of annealing treatment after arc-melting? Is the high background observed at low angle due to the holder or does it originate from the possible amorphization induced by ball-milling? Some samples further show very broad humps instead of neat peaks (especially in Figures 4 an 6). What do they correspond to?

Authors mentioned the apparition of secondary phases and in some cases of FCC while the as-prepared alloys are BCC. As you know, some BCC medium/high entropy alloys undergo a phase transformation from BCC to FCC upon hydride formation, which is a reversible process upon hydrogen release. Why has this not been discussed nor  compared with the existing literature?

Finally, the high signal/noise ratio makes it difficult to clearly differentiate and visualize as-prepared versus after hydrogen charging curves. The Authors are invited to revise the figures to improve clarity.

Author Response

Dear reviewer,

Thank you very much for the comments and recommendations, I am sending the edited article in the attachment.

Best regards 

Varcholova et al.

Reviewer 2 Report

Comments and Suggestions for Authors

The manuscript "New- generation materials for hydrogen storage in medium-entropy alloys", submitted for publication on Materials, has been reviewed. It deals with the design, preparation and characterization of new medium-entropy alloys for hydrogen storage. Three systems have been analyzed (Al-Ti-Nb-Zr, Al-Ti-Nb-V, Al-Ti-Nb-Hf) without rare earth elements. Experimental results show a strong correlation between hardness and hydrogen absorption/desorption.

The manuscript is clear, well arranged and relatively novel. English is fine. Results are almost clearly presented, conclusions supported by the results. After an introduction on the future targets of EU for energy and climate strategy, the manuscript presents results of nanoindentation tests, XRD, absorption/desorption capacity and cycle efficiency.

However, in my opinion, it can be accepted after the following major revisions:

1) Line 99: Vickers not Wicher.

2) Line 133: please specify X-ray wavelength. 

3) Text in all figures are too small and unreadable. Please enlarge or increase font size.

After that it can be reconsidered for publication on Materials.

Author Response

Dear reviewer,

Thank you very much for the comments and recommendations, I am sending the edited article in the attachment.

Best regards 

Reviewer 3 Report

Comments and Suggestions for Authors

Dear authors

I have some questions and comments about the article.

My questions and comments:

1.      The introduction should be expanded with current articles related to hydrogen storage in high-entropic materials.

2.      Figures 1, 2, and 3 are not very legible and should be corrected

3.      In many places in the text, the chemical formulas of compounds should be corrected so that they have subscripts and not simple numbers

4.      Referring to Table 1,2,3, how do microhardness, nano hardness, and elastic modulus depend on changes in the alloy composition?

5.      In Figures 4, 5, and 6 we have XRD spectra of the basic material and after hydrogenation, why do we not observe a shift of the XRD peaks to lower angles after hydrogenation in each spectrum?

6.      As for tables 4, 5, and 6, how do S, VEC, absorption, residual, desorption, and cycle efficiency depend on changes in the composition of compounds?

7.      As for Figures 7, 8, 9, and 10, they are not very legible, they should be corrected

8.      The article is more like a report of the results, there is not much scientific analysis of the results obtained, and little is clear from the conclusions.

Comments on the Quality of English Language

Dear authors

I have some questions and comments about the article.

My questions and comments:

1.      The introduction should be expanded with current articles related to hydrogen storage in high-entropic materials.

2.      Figures 1, 2, and 3 are not very legible and should be corrected

3.      In many places in the text, the chemical formulas of compounds should be corrected so that they have subscripts and not simple numbers

4.      Referring to Table 1,2,3, how do microhardness, nano hardness, and elastic modulus depend on changes in the alloy composition?

5.      In Figures 4, 5, and 6 we have XRD spectra of the basic material and after hydrogenation, why do we not observe a shift of the XRD peaks to lower angles after hydrogenation in each spectrum?

6.      As for tables 4, 5, and 6, how do S, VEC, absorption, residual, desorption, and cycle efficiency depend on changes in the composition of compounds?

7.      As for Figures 7, 8, 9, and 10, they are not very legible, they should be corrected

8.      The article is more like a report of the results, there is not much scientific analysis of the results obtained, and little is clear from the conclusions.

Author Response

(The authors gave the same response as above.)

Reviewer 4 Report

Comments and Suggestions for Authors

The study presents a exploration into medium-entropy alloys for hydrogen storage. this work contributes to the broader goal of sustainable energy solutions by improving the efficiency of hydrogen storage without relying on rare earth materials. The paper is in the scope of the journal. However, there are few comments to consider. Accordingly, I would recommend publishing the article after addressing the following comments and concerns:

-          The paper mentions various applications of hydrogen. However, the literature review on medium-entropy alloys (MEAs) specifically related to hydrogen storage is limited. Please expand it

-          The introduction provides an overview of hydrogen’s role in energy and climate strategy. However, it could focus on discussion on the specific challenges in hydrogen economy. You can use the following paper as reference: doi.org/10.1039/D3CS00723E

-          The objective of designing new alloys for improved hydrogen storage is mentioned, but a clear statement of research questions or hypotheses better to be added within the initial sections.

-          Please add a discussion on rationale behind choosing specific elemental compositions.

-          Maybe also mention other options of hydrogen storage materials, please note clathrates hydrates, you can use following reference: doi.org/10.1039/C8CS00989A

-          Please add details on reasoning for the chosen methods of hydrogenation and characterization.

-          Please add what did you do for repeatability and also discuss the details of any control experiments or how you controlled the conditions.

-          Some of the text in figures are hard to read, please make sure all are readable font.

-          The paper mentions standard deviations. However, a comprehensive statistical analysis of the data variability is missing.

-          The results section clearly presents the obtained data . However, there is no deeper analysis comparing the performance of the alloys beyond just the raw data. Maybe it can be improved through normalized comparisons or percentage improvements.

-          Have you done any analysis on long term stability of this storage options?

Author Response

(The authors gave the same response as above.)

Round 2

Reviewer 2 Report

Comments and Suggestions for Authors

The improved manuscript can be accepted as it is

Author Response

Dear Reviewer,

we want to express our sincere gratitude for your diligent and constructive review of our manuscript. Your valuable insights and feedback have played a crucial role in enhancing the quality of our work.

Sincerely,

Varcholová et al.

Reviewer 3 Report

Comments and Suggestions for Authors

Dear authors

I have some questions and comments about the article.

My questions and comments:

1.      Figures 1, 2, and 3 are not very legible and should be corrected

2.      Referring to Table 1,2,3, how do microhardness, nano hardness, and elastic modulus depend on changes in the alloy composition?

3.      In Figures 4, 5, and 6 we have XRD spectra of the basic material and after hydrogenation, why do we not observe a shift of the XRD peaks to lower angles after hydrogenation in each spectrum?

4.      As for tables 4, 5, and 6, how do S, VEC, absorption, residual, desorption, and cycle efficiency depend on changes in the composition of compounds?

5.      As for Figures 7, 8, 9, and 10, they are not very legible, they should be corrected

6.      The article is more like a report of the results, there is not much scientific analysis of the results obtained, and little is clear from the conclusions.

Author Response

Dear Reviewer,

we extend our heartfelt appreciation for the diligent and constructive review of our manuscript. Your valuable insights and feedback have significantly contributed to the improvement of our work. In response to your comments, we have carefully addressed each point, providing detailed explanations and incorporating necessary revisions. The attached response document outlines the specific changes made, with revised sections highlighted in green for your convenience. Your commitment to maintaining the scholarly rigor of our manuscript has been instrumental, and we believe that these enhancements have strengthened the overall quality and clarity of our work. We are truly grateful for your time, expertise, and thoughtful contributions. Thank you once again for your invaluable assistance in shaping our manuscript into a more robust and refined piece of research.

Sincerely,

Varcholová et al.
